# The impact of atlas-based MR attenuation correction on the diagnosis of FDG-PET/MR for Alzheimer's diseases— A simulation study combining multi-center data and ADNI-data

**Tetsuro Sekine**[1,2]*, **Alfred Buck**[1,3], **Gaspar Delso**[4], **Bradley Kemp**[5], **Edwin E. G. W. ter Voert**[1,3], **Martin Huellner**[1,3], **Patrick Veit-Haibach**[1,3,6,7], **Sandeep Kaushik**[4], **Florian Wiesinger**[4], **Geoffrey Warnock**[8,9], **for the Alzheimer's Disease Neuroimaging Initiative**[¶]

**1** Department of Nuclear Medicine, University Hospital Zurich, Zurich, Switzerland, **2** Department of Radiology, Nippon Medical School, Tokyo, Japan, **3** University of Zurich, Zurich, Switzerland, **4** GE Healthcare, Waukesha, WI, United States of America, **5** Department of Radiology, Mayo Clinic, Rochester, MN, United States of America, **6** Department of Medical Radiology, Division of Diagnostic and Interventional Radiology, University Hospital Zurich, Zurich, Switzerland, **7** Department Joint Medical Imaging, Toronto General Hospital, Toronto, ON, Canada, **8** Institute of Pharmacology & Toxicology, University of Zurich, Zurich, Switzerland, **9** PMOD Technologies Ltd., Zürich, Switzerland

¶ See Acknowledgments for more information.
* tetsuro.sekine@gmail.com

**Data Availability Statement:** ADNI-PET data is available on ADNI website (http://adni.loni.usc.edu/). There are ethical restrictions on sharing

## Abstract

### Background

The purpose of this study was to assess the impact of vendor-provided atlas-based MRAC on FDG PET/MR for the evaluation of Alzheimer's disease (AD) by using simulated images.

### Methods

We recruited 47 patients, from two institutions, who underwent PET/CT and PET/MR (GE SIGNA) examination for oncological staging. From the PET raw data acquired on PET/MR, two FDG-PET series were generated, using vendor-provided MRAC (atlas-based) and CTAC. The following simulation steps were performed in MNI space: After spatial normalization and smoothing of the PET datasets, we calculated the error map for each patient, $PET_{MRAC}/PET_{CTAC}$. We multiplied each of these 47 error maps with each of the 203 Alzheimer's Disease Neuroimaging Initiative (ADNI) cases after the identical normalization and smoothing. This resulted in 203*47 = 9541 datasets. To evaluate the probability of AD in each resulting image, a cumulative t-value was calculated automatically using commercially-available software (PMOD PALZ) which has been used in multiple large cohort studies. The diagnostic accuracy for the discrimination of AD and predicting progression from mild cognitive impairment (MCI) to AD were evaluated in simulated images compared with ADNI original images.

patients's dataset because it contains potentially identifying or sensitive patient information. The authors had no special access privileges to the data others would not have. It means that any interested researchers can replicate the current study findings in their entirety by directly obtaining the data from ADNI's website and combining their own error map dataset.

**Funding:** This study was funded by: TS received investigator initiated study grants from Hitachi Global Foundation, Fukuda Foundation for Medical Technology and Terumo foundation for life sciences and arts. PVH received IIS grants from Bayer Healthcare, Roche Pharmaceutical, GE Healthcare and Siemens Healthcare. GW was funded by the Clinical Research Priority Program for Molecular Imaging of the University of Zurich (MINZ).GE Healthcare provided support for this study in the form of salaries for: G.D., S.K., F.W. PMOD Technologies LLC provided support for this study in the form of a salary for GW.The specific roles of these authors are articulated in the 'author contributions' section. This work was supported by JSPS KAKENHI (Grant Number 17K18160), Kurata Grants from the Hitachi Global Foundation (Grant Number 1309), research grants from Fukuda Foundation for Medical Technology and Terumo foundation for life sciences and arts. Data collection and sharing for this project was funded by the Alzheimer's Disease Neuroimaging Initiative (ADNI) (National Institutes of Health Grant U01 AG024904) and DOD ADNI (Department of Defense award numberW81XWH-12-2-0012).Data collection and sharing for this project was funded by the Alzheimer's Disease Neuroimaging Initiative (ADNI) (National Institutes of Health Grant U01 AG024904)and DOD ADNI (Department of Defense award number W81XWH-12-2-0012). "The other funders had no role in study design, data collection and analysis, decision to publish, or preparation of the manuscript. "ADNI had no additional role in study design and data analysis, decision to publish, or preparation of the manuscript."

**Competing interests:** The authors have read the journal's policy and the authors of this manuscript have the following competing interests: TS received investigator initiated study grants from Hitachi Global Foundation, Fukuda Foundation for Medical Technology and Terumo foundation for life sciences and arts. PVH received IIS grants from Bayer Healthcare, Roche Pharmaceutical, GE Healthcare and Siemens Healthcare, and speaker fees from GE Healthcare. Three authors (G.D., S.K., F.W.) are paid employees of GE Healthcare. Only non-GE employees had control of inclusion of data and information that might present a conflict of

## Results

The accuracy and specificity for the discrimination of AD-patients from normal controls were not substantially impaired, but sensitivity was slightly impaired in 5 out of 47 datasets (original vs. error; 83.2% [CI 75.0%-89.0%], 83.3% [CI 74.2%-89.8%] and 83.1% [CI 75.6%-88.3%] vs. 82.7% [range 80.4–85.0%], 78.5% [range 72.9–83.3%,] and 86.1% [range 81.4–89.8%]). The accuracy, sensitivity and specificity for predicting progression from MCI to AD during 2-year follow-up was not impaired (original vs. error; 62.5% [CI 53.3%-69.3%], 78.8% [CI 65.4%-88.6%] and 54.0% [CI 47.0%-69.1%] vs. 64.8% [range 61.5–66.7%], 75.7% [range 66.7–81.8%,] and 59.0% [range 50.8–63.5%]). The worst 3 error maps show a tendency towards underestimation of PET scores.

## Conclusion

FDG-PET/MR based on atlas-based MR attenuation correction showed similar diagnostic accuracy to the CT-based method for the diagnosis of AD and the prediction of progression of MCI to AD using commercially-available software, although with a minor reduction in sensitivity.

## Background

Integrated positron emission tomography (PET) / magnetic resonance (MR) systems have been currently widely distributed (over 100 institutions in the world). Previous studies have revealed that 2-deoxy-2-[18F]fluoro-D-glucose (FDG)-PET/MR is useful in the evaluation of neurodegenerative diseases [1–6]. Additionally, combined PET/MR not only provides detailed brain anatomy, but the immediate availability of coregistered anatomy might even improve PET image quality by facilitating the correction of partial volume effects and/or motion artifacts [7, 8]. However, several technical challenges should be solved to exploit the full performance of PET/MR. One of the limitations in need of improvement is that of attenuation correction (AC) using MR imaging data (MRAC) [9]. On PET/MR system, it is difficult to derive AC-maps from conventional MR-data due to the lack of a relationship between photon attenuation and MR signal intensity. To solve this problem, several AC-methods (i.e. Dixon-based AC, Atlas-based AC, Model-based AC, zero echo time MRI based AC and ultrashort echo time MRI based AC) have been proposed from vendors and researchers [9, 10]. For clinical use, Dixon-based four-class segmentation approaches (i.e. air, lung, fat and soft tissue) was implemented into both the Biograph mMR (Siemens Healthcare, Erlangen, Germany) and SIGNA PET/MR (GE Healthcare, Waukesha, WI, USA). However, these methods are not recommended for brain studies, because neglecting bone introduces a significant bias in cortical areas [11]. One of the alternative methods currently implemented on clinical PET/MR scanners is the atlas-based method [12, 13]. This method is comparably accurate in supratentorial regions, but not accurate enough in the temporal lobe and in the infratentorial region, where FDG uptake is underestimated. This variability of error distribution may impact the diagnostic accuracy of FDG PET in several diseases, including Alzheimer's disease (AD). Quantitative evaluation of FDG-PET typically relies on a normalization of local FDG uptake to that in dedicated anatomical regions (e.g. cerebellum and thalamus) or to the whole brain average. If the regions with overestimated FDG accumulation are normalized to an underestimated region, or vice versa, the result could be under- or over-diagnosis of AD. However, the impact of the

interest for authors who are employees of GE Healthcare. GW is a paid employee of PMOD Technologies LLC. There are no patents, products in development or marketed products associated with this research to declare. This does not alter our adherence to PLOS ONE policies on sharing data and materials. No other potential conflicts of interest relevant to this article exist.

**Abbreviations:** AC, Attenuation correction; AD, Alzheimer's disease; ADNI, Alzheimer Disease Neuroimaging Initiative; CI, Confidence interval; FDG, 2-deoxy-2-[18F]fluoro-D-glucose; LAVA-Flex, liver acquisition with volume acceleration flex; MCI, Mild cognitive impairment; MR, Magnetic resonance; MRI, Magnetic resonance imaging; PET, Positron emission tomography.

vendor-provided MRAC on the accuracy of diagnosis of AD has not been reported in the literature.

The aim of this paper was to clarify the clinical utility of FDG-PET from PET/MR, with vendor-provided atlas-based MRAC, for the diagnosis of AD. The analysis was performed on simulated data that combined real patient data from two institutions (Institution A (InA) and B (InB)) and Alzheimer Disease Neuroimaging Initiative (ADNI) data, well-established large cohort data. The probability of AD was calculated using fully automated procedures in commercially available software.

## Materials and methods

This study was approved by each local institutional review board, cantonal ethics committee Zurich and Institutional Review Board of Mayo Clinic. All subjects provided signed informed consent prior to the examinations. All experiments were performed in accordance with relevant guidelines and regulations. We recruited 47 patients, from two institutions, who underwent both PET/CT and PET/MR (GE SIGNA) examination. In addition, we extracted 203 subjects from the ADNI dataset. From the PET raw data acquired on PET/MR, two FDG-PET series were generated, using either the vendor-provided atlas-based MRAC or CTAC. Following spatial normalization to MNI space, we calculated the error map for each patient, as $PET_{MRAC}/PET_{CTAC}$. We multiplied each of these 47 error maps with each of the 203 ADNI cases in MNI space. This resulted in 203*47 = 9541 datasets (**Fig 1**). To evaluate the probability of AD in each resulting image, a cumulative t-value was calculated using a fully automated method in commercially-available software (**S1 Fig**). The diagnostic accuracy for the discrimination of AD and predicting progression from mild cognitive impairment (MCI) to AD were evaluated in simulated images compared with the original ADNI images (**Fig 1**).

### Alzheimer Disease Neuroimaging Initiative (ADNI) data

Data used in the preparation of this article were obtained from the ADNI database (adni.loni. usc.edu). ADNI was launched in 2003 as a public-private partnership, led by Principal Investigator Michael W. Weiner, MD. The primary goal of ADNI has been to test whether serial magnetic resonance imaging (MRI), positron emission tomography (PET), other biological markers, and clinical and neuropsychological assessment can be combined to measure the progression of mild cognitive impairment (MCI) and early Alzheimer's disease (AD). For up-to-date information, see www.adni-info.org. The authors had no special access privileges to the data others would not have. It means that any interested researchers can replicate the current study findings in their entirety by directly obtaining the data from ADNI's website and combining their own error map dataset.

From ADNI-1 data, we extracted 203 participants (76.0±6.3 years, 129 males, 48 healthy, 59 AD and 96 MCI participants). Out of 96 MCI participants, 33 progressed to AD at 24 months after imaging. The inclusion criteria were: completeness of date of birth, baseline diagnosis (healthy, MCI, or AD) and the diagnosis at 24 months after imaging. All PET images were of sufficient quality for visual scoring and for software-based analysis using PMOD's Alzheimer Discrimination tool PALZ (PMOD Technologies LLC, Zurich, Switzerland) [14, 15]. The baseline PET data was utilized. The reported FDG-PET imaging parameters were: injected dose, 185 MBq (5 mCi), dynamic 3D acquisition, six 5-min frames 30–60 min post injection.

### Patients

We recruited 47 patients who underwent both PET/CT and PET/MR for oncologic staging from two institutions (InA and InB). Twenty patients (11 males and 9 females, 61.6±12.4

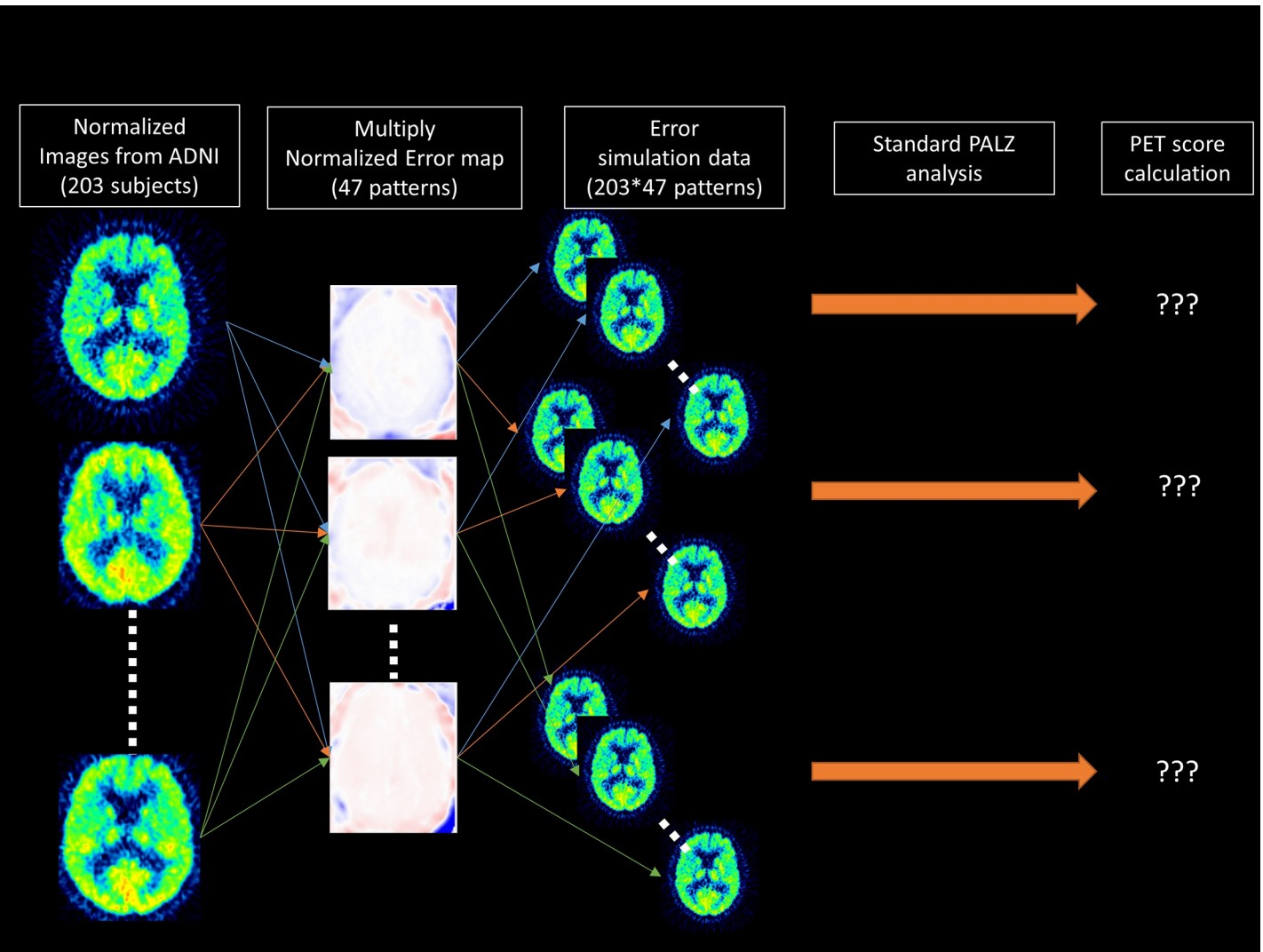

**Fig 1. Summarized workflow used to generate the simulated PET dataset with error derived from MRAC.** A detailed description of the procedure is provided in the materials and methods section.

years) with lymphoma, lung cancer, head and neck cancer, pancreatic cancer, pleural methotelioma, uterine carcinoma, cervical carcinoma and malignant melanoma at InA were collected by summing the cohorts recruited in previous studies [12, 16]. The other twenty-seven patients (15 males and 12 females, 60.0±13.0 years) with lymphoma, pheochromocytoma, myeloma, melanoma, Merker-cell cancer, lung cancer, pancreatic cancer, breast cancer and dementia at InB were collected from another previous study, after excluding 3 patients. Two of these three excluded patients had infarction and one had multiple brain metastases [17]. A neuroradiologist (T.S.) reviewed and confirmed that all included patients were free of brain abnormalities.

## PET/CT and PET/MR acquisition

The averaged injected dose of FDG was 233.2±42.1MBq (range, 179.4–325.7 MBq) at InA and 534±42 MBq [range, 434–566 MBq] at InB. The PET/CT acquisition followed the standard protocol for a clinical oncology study. InA used a Discovery 690 PET/CT (GE Healthcare) and

InB used a Discovery RX/MI/690/710 PET/CT (GE Healthcare). A helical whole-body CT scan (120 kV, slice thickness 3.3mm, pixel dimensions 1.4×1.4 mm$^2$ at InA and 120–140 kV, slice thickness 3.75–5.00 mm, pixel size 1.37×1.37 mm$^2$ at InB) was acquired for AC of PET data and diagnostic purposes [18]. Subsequently, a whole-body PET dataset including the head was acquired. Immediately before or after the PET/CT scan, patients were transferred to the integrated PET/MR scanner (SIGNA PET/MR, GE Healthcare), and a brain PET/MR scan was performed as part of the study examination. A 2 min PET acquisition at InA and a 10 min acquisition at InB with a standard head coil (8-channnel HD Brain; GE Healthcare) was performed. The duration between tracer injection and PET acquisition was 116±31 min [range, 48–183 min] at InA and 112±15 min [range, 66–138 min] at InB.

During the PET acquisition on the PET/MR, liver acquisition with volume acceleration flex (LAVA-Flex) T1w images (axial acquisition, TR ~ 4 ms, TE 2.23 ms, flip angle 5˚, slice thickness 5.2 mm with 2.6 mm overlap, 120 slices, pixel size 1.95 × 1.95 mm$^2$, number of excitations (NEX) 0.9, acquisition time: 18 s) were acquired for vendor-provided atlas-based AC [13].

## Attenuation map generation

The atlas AC map was calculated from the LAVA-Flex T1w images using the vendor-provided default processing [18].

For the generation of CTAC, the processing steps detailed below were performed using custom Matlab scripts and PMOD 3.8. The co-registered CTAC map was generated as follows. First, the original head CT was exported from the PET/CT scanner and converted into an AC map using a Matlab version of the same bilinear mapping implemented in the SIGNA PET/MRI. From this map, the CT table was removed manually. A threshold was set to extract the outside air component from the CTAC map. None of the images used in this study contained artifacts likely to affect air thresholding. To derive the registration parameters necessary to match CT to LAVA-Flex T1w, a normalized mutual information matching algorithm (PMOD) was used and the final matching was performed using custom Matlab routines. Finally, the CTAC map was superimposed on the atlas AC map, replacing it [13].

## Reconstruction of PET images

Only the list-mode raw PET data from the PET/MR examination were used. PET images were reconstructed twice, using either atlas AC or silver-standard CTAC using the following parameters: InA, fully 3D ordered subset expectation maximization iterative reconstruction (OSEM), subsets 28, iterations 8, pixel size 1.17 × 1.17 mm$^2$, point spread function (PSF) modeling, transaxial post-reconstruction Gaussian filter cutoff 3mm, axial filter 1:4:1, scatter, normalization, dead-time and decay corrections, TOF reconstruction; InB, fully 3D OSEM, subsets 28, iterations 3, pixel size 1.56 × 1.56 mm$^2$, PSF modeling, transaxial post-reconstruction Gaussian filter cutoff 3mm, axial filter 1:4:1, scatter, normalization, dead-time and decay corrections, TOF reconstruction.

## Automated software for AD probability assessment

Automated AD probability assessment was performed in a commercially available tool (PMOD Alzheimer's Discrimination, PALZ). This software tool has been used in multiple large cohort studies, e.g. ADNI, NEST-DD and SEAD-Japan [15, 19, 20]. The software ran the following procedures, in a fully automated workflow, in accordance to the methods described by Herholz et al. [21]. First, spatial normalization is performed by transforming the original images to the SPM99 PET template, followed by smoothing with a Gaussian filter of 12 x 12 x 12 mm [22, 23]. In these images, voxel values are normalized by dividing each image voxel

value by the mean voxel value, averaged within a mask representing voxels in which FDG uptake is typically preserved even in AD patients. The expected value in each voxel is calculated from a pre-stored, age-matched, reference PET database of healthy controls. This is achieved by combining voxel-wise regression parameters, where brain atrophy was taken into account by adjustment of the normal reference values using linear regression by age. By comparing the voxel-wise differences between expected value and the patient-specific value, a Student's t-value is calculated [24]. The AD t-sum is calculated by summing the t-value in predefined AD-related voxels. Finally, the PET Score was calculated as log2 (AD t-sum/11089 +1), for which the 95% prediction limit (11089) of AD t-sum was established in the NEST-DD multi-center trial [15]. This analysis was initially performed in all 203 ADNI-PET data (e.g. $PETScore_{ADNI-j}$) before multiplication with the 47 error maps. The detailed procedure is shown in S1 Fig.

## Creation of simulated data: ADNI-data with Atlas-AC

All simulation steps were performed in MNI space with same spatial resolution (2 mm isotropic voxels). First, we divided the locally acquired PET images based on atlas AC by those based on CTAC (47 patients) (e.g. $_{Error}PET^{pt-i}$). Second, the resulting images were spatially normalized to the SPM99 PET template using the transformation calculated for PET images based on CTAC to the template, then a Gaussian filter of 12 x 12 x 12 mm full-width half-maximum was applied ($_{Error}^{Norm}PET^{pt-i}$). A brain mask was applied to avoid distortion at the edges of the measured data. These steps were designed to replicate the preprocessing steps used in the PMOD Alzheimer's Discrimination tool, as used to calculate PET score. Therefore, the resulting images were the error maps (between atlas AC and CTAC) in the same image space as the spatially normalized ADNI PET data ($^{Norm}PET_{ADNI-j}$). Third, we multiplied each of the 203 normalized ADNI data with each of the 47 normalized error maps, resulting in 203*47 = 9541 normalized PET images (e.g. $_{Error}^{Norm}PET_{ADNI-j}^{pt-i}$). Thus, the value-error was simply imposed in a voxel-wise manner and further PET score calculation was performed without additional need for spatial deformation or filtering. Therefore, we expected any bias due to impaired spatial normalization or differences in PET acquisition protocol to be minimized. For each of these 9541 images, PALZ analysis (from the second to fifth analysis step) was performed to calculate the PET score ($PETScore_{ADNI-j}^{pt-i}$). This workflow is summarized in **Fig 1**.

## Evaluation of diagnostic accuracy for Alzheimer's disease

First, to clarify the distribution of MRAC error, we calculated the averaged error in whole SPM 99 PET voxels, whole AD-related voxels and whole non-AD related voxels in each of the 47 normalized error map ($_{Error}^{Norm}PET^{pt-i}$). Second, we calculated the difference in PET score ($PETScore_{ADNI-j}^{pt-i} - PETScore_{ADNI-j}$) for all 9541 datasets. To reveal whether differing PET acquisition protocols affected the simulation results, we additionally compared PET score between InA and InB.

We evaluated the diagnostic accuracy from two points of view. First was the diagnostic accuracy of discrimination of AD from normal patients. Based on a previous study, the cut-off is PET score = 1 [15]. Second was the diagnostic accuracy of prediction of conversion from MCI to AD [14]. For the prediction from MCI to AD, the PET score cut-off is 0.79, defined using the Youden index [25].

We calculated sensitivity and specificity using the data for each error map multiplied by each PET image. We created Bland-Altman plots of PET scores in the best 3 and the worst 3 cases [26]. The confidence intervals (CI) were calculated using the original data.

**Table 1. Diagnostic accuracy of original PET data and simulated MRAC PET data.**

| | | Original | MRAC (47 datasets) | | |
|---|---|---|---|---|---|
| | | | Average | Worst | Best |
| **Discrimination AD from normal** | Accuracy | 83.2% (CI: 75.0–89.0%) | 82.7% | 80.4% | 85.0% |
| | Sensitivity | 83.3% (CI: 74.2–89.8%) | 78.5% | 72.9% | 83.3% |
| | Specificity | 83.1% (CI 75.6–88.3%) | 86.1% | 81.4% | 89.8% |
| **Prediction of progression from MCI to AD** | Accuracy | 62.5% (CI: 53.3–69.3%) | 64.8% | 61.5% | 66.7% |
| | Sensitivity | 78.8% (CI: 65.4–88.6%) | 75.7% | 66.7% | 81.8% |
| | Specificity | 54.0% (CI: 47.0–69.1%) | 59.0% | 50.8% | 63.5% |

AD, Alzheimer's disease; MCI, mild cognitive impairment; MRAC, MR-based attenuation correction.

## Results

In 47 normalized error map, whole voxels of SPM 99 were slightly underestimated (-1.37% ±1.98%). In detail, AD-related voxels were less underestimated than non AD-related voxels (-0.86%±2.03% vs. -1.59%±1.98%). As a result, compared with the reference PET score derived from original 203 ADNI data, the PET score from 9541 simulated data was slightly underestimated (-0.068±0.046). There was no statistical difference in PET score between the patient groups from InA and InB (-0.070±0.045 vs. -0.066±0.046; p = 0.392).

Neither accuracy nor specificity for the discrimination of AD patients from normal controls were significantly impaired by MRAC, but sensitivity was slightly impaired when using 5 out of 47 error maps (original vs. error; 83.2% [CI 75.0-89.0%], 83.3% [CI 74.2-89.8%] and 83.1% [CI 75.6-88.3%] vs. 82.7% [range 80.4–85.0%], 78.5% [range 72.9–83.3%,] and 86.1% [range 81.4–89.8%]) (**Table 1** and **Fig** 2A).

The accuracy, sensitivity, or specificity for predicting progression from MCI to AD during 2-year follow-up were maintained by MRAC (62.5% [CI 53.3-69.3%], 78.8% [CI 65.4-88.6%] and 54.0% [CI 47.0-69.1%] vs. 64.8% [range 61.5–66.7%], 75.7% [range 66.7–81.8%,] and 59.0% [range 50.8–63.5%]) (**Table 1** and **Fig 2B**). The worst 3 error maps showed a tendency towards underestimation of PET scores (**Fig 3**). A representative case is shown in **Fig 4**.

## Discussion

In the current study, we estimated the diagnostic accuracy of FDG-PET with vendor-provided atlas AC from PET/MR (GE SIGNA) for AD. This was achieved by simulating the error introduced by MRAC on ADNI data and investigating the subsequent effect on an automated method for Alzheimer's discrimination. The result shows that error induced by MRAC could lead to an underestimation of the probability of AD. Accuracy and specificity were maintained, but sensitivity for the discrimination of Alzheimer's disease from normal subjects was slightly impaired. A similar slight tendency was found for the prediction of progression from MCI to AD.

There have been few studies evaluating diagnostic accuracy for AD in clinical PET/MR machines that included more than 10 patients. Hitz et al. recruited 30 patients with suspected AD [27]. FDG-PET imaging on PET/CT with CTAC and that on PET/MR with MRAC were generated separately. Quantitative analysis showed that inconsistent over- and underestimation, depending on the anatomical region, was apparent on PET/MR even after normalization to the global mean. In visual assessment, even experienced observer ratings diverged between PET/CT and PET/MR in 3 out of 29 patients. In this study, they used Dixon-based MRAC, which is no longer recommended for use in brain PET/MR imaging, and generated PET data

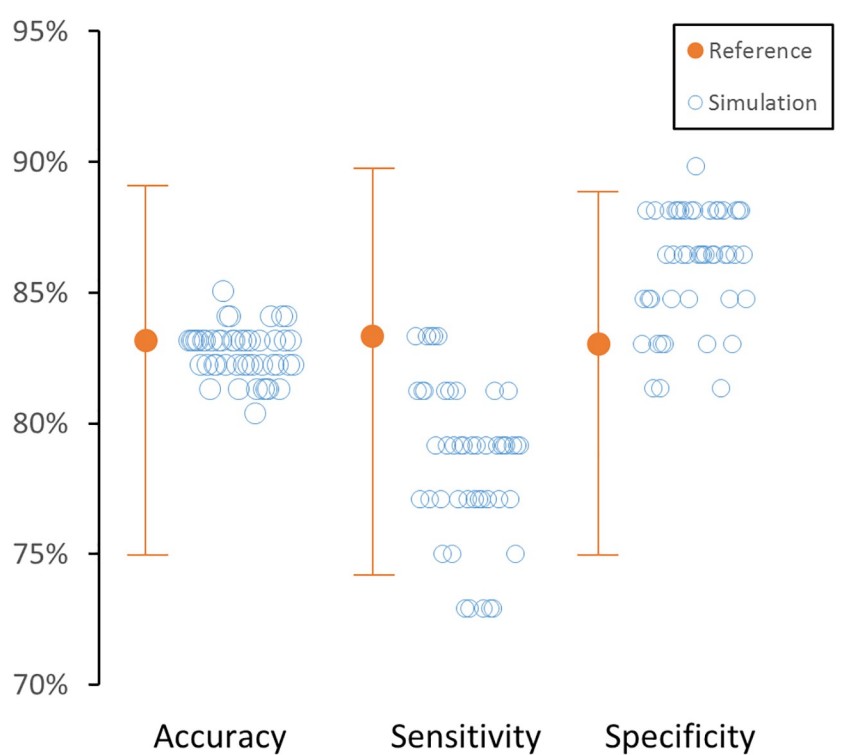

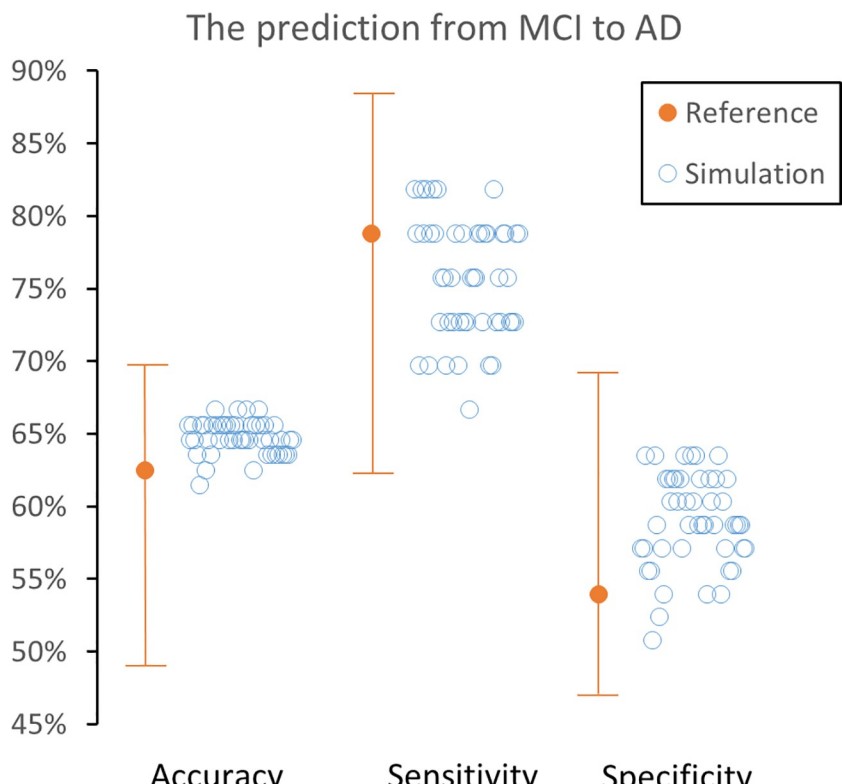

**Fig 2. Accuracy, sensitivity and specificity in original ADNI data and each of the 47 datasets with simulated MRAC error.** Discrimination of Alzheimer's disease (AD) patients from normal subjects (A) and the prediction of mild cognitive impairment (MCI) progression to AD. 95% confidence intervals are shown as Whisker plots. All 47 datasets maintain their accuracy and specificity compared with the original data, while five out of 47 error datasets slightly impaired the sensitivity for the discrimination of AD.

for each modality from different PET raw data [28]. Our goal was to evaluate the vendor-provided atlas AC, and to use the same PET raw data in the generation of error maps.

Moodley et al. enrolled 24 dementia patients [29]. However, the main focus of their study was to evaluate the concordance between FDG-PET and MRI in dementia patients, rather than the validation of MRAC compared with the gold standard: FDG-PET derived from CTAC.

Further studies have been published on the validation of MRAC for brain FDG-PET, some of which recruited AD patients [10, 30, 31]. However, the main focus of these studies was to clarify the extent and distribution of error introduced by MRAC. None of these studies evaluated the impact of these errors on the diagnostic accuracy for AD in an objective manner. In addition, these studies were performed on different MR systems with different underlying

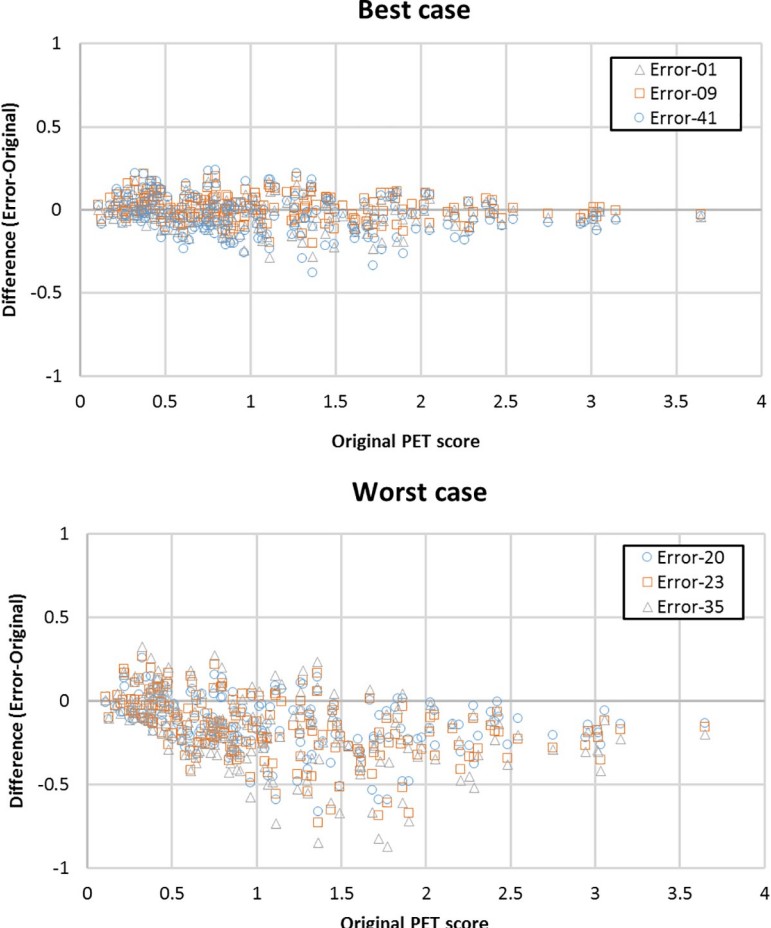

**Fig 3.** Bland-Altman plots of best three cases (A) and worst three cases (B) for 203 simulated PET data with MRAC error. The PET score tends to be underestimated, especially in the worst cases.

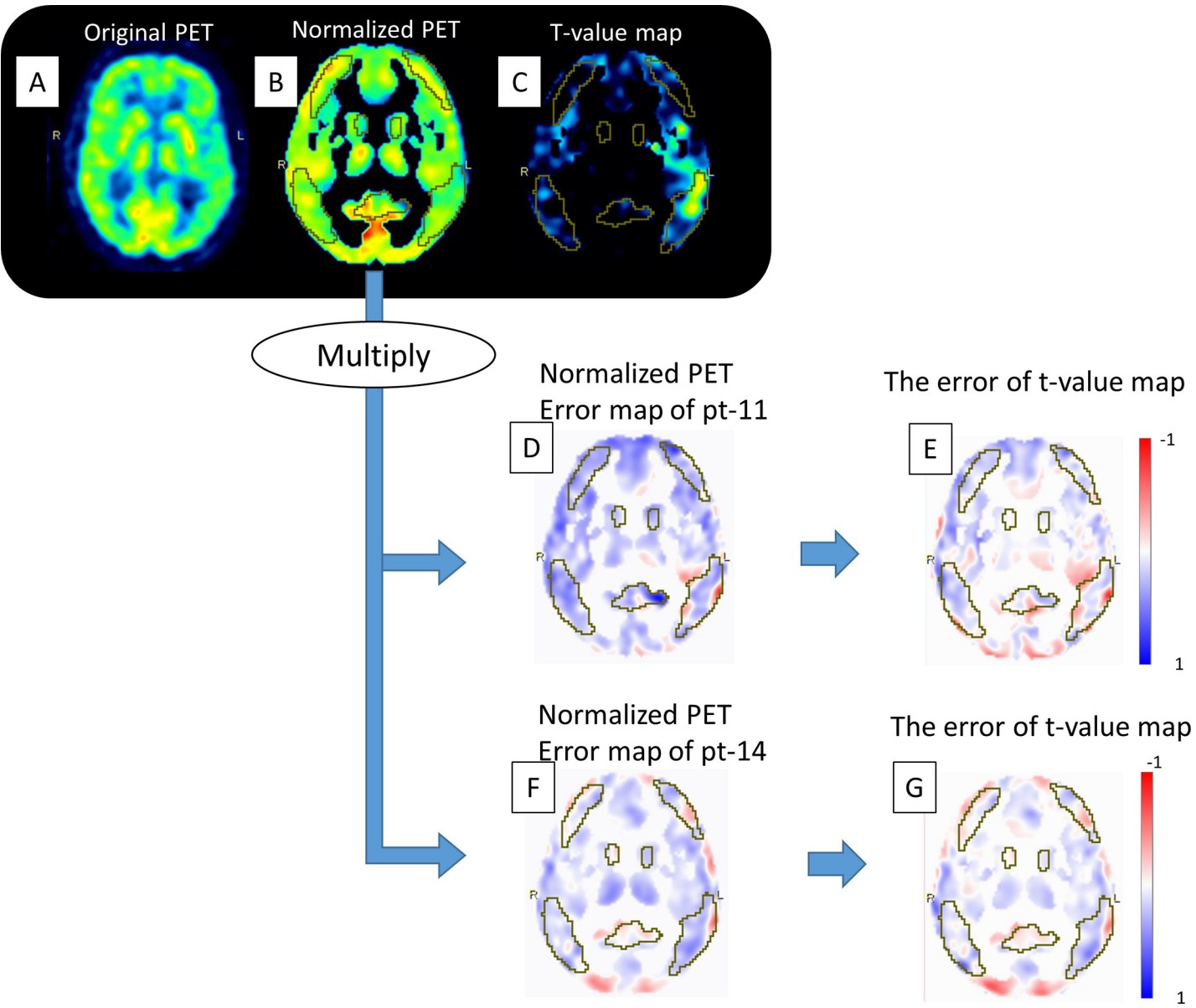

**Fig 4. Representative case.** The original PET images extracted from ADNI-cohort (A). After spatial and value normalization, the normalized images were acquired (B). The voxel-wise differences between the expected and patient-specific images are used to calculate t-values (C). The original error map was obtained by dividing PET based on Atlas-AC by that on CT-AC, followed by normalization identical to that used for the ADNI-PET data (D, F). We multiplied the normalized PET (B) with each normalized error map (D and F), followed by calculation of the t-value map of the simulated PET (not shown in this figure). The error of the t-value maps was obtained by subtracting the original t-value map from the simulated t-value map (E and G). More detailed figures are shown in S2 Fig.

PET-technology, i.e. the Siemens mMR. There is evidence that the time-of-flight (TOF) capability of the GE SIGNA PET/MR might compensate for errors introduced by MRAC [12].

We used a fully automated, commercially available, Alzheimer's discrimination analysis to calculate a predictive value for AD. The normal database of FDG PET images used for the discrimination (calculation of t-value) in this tool were acquired on conventional PET/CT scanners [15]. Based on our study alone, it is difficult to determine whether a PET/MR-specific FDG-PET database should be constructed [27]. However, our finding that diagnostic accuracy

was maintained between FDG-PET/MR with atlas-based MRAC and PET/CT indicates that the database for FDG-PET/CT could be used for FDG-PET/MR.

A recent multi-center trial revealed that some novel MRAC methods could generate near "gold standard" AC-maps from MR data on PET/MR [30]. In addition, deep-learning-based MRAC methods have been validated and provided prominent accuracy [32–34]. However, implementation and adoption on commercially available systems will certainly vary among vendors and users. At most institutions with a PET/MR scanner, researchers and physicians can only choose between the MRAC provided by the vendor, a multi-atlas based method available via a web interface (http://cmictig.cs.ucl.ac.uk/niftyweb/program.php?p=PCT) or their own reconstruction algorithm [12, 35].

There were a number of limitations in our study. First, the data acquisition in patients was heterogeneous and not optimized in all cases for brain PET (short 2 min scan duration and variable post-injection time). Using only 2 min acquisition for brain PET images could limit image quality, despite the use of a state-of-the-art PET/MR scanner with high sensitivity detector [36]. To mitigate this, all images were smoothed with a Gaussian filter of 12 x 12 x 12 mm which step has been included in PALZ as one of the analysis steps. In addition, we proved that there was no significant difference of PET score error between short scan time (2min) cohort at InA and relatively longer scan time (10min) cohort at InB. Second, brain PET data was taken from patients imaged for oncologic staging, rather than dedicated brain imaging. However, the error introduced by MRAC primarily results from differences in skull bone detection which is not expected to systematically vary among oncology and AD patients. Third, a limited number of patients, n = 47, were used to generate MRAC error maps. This is substantially smaller than a previous larger cohort study that validated the performance of MRAC (n = 337) [30]. In addition, the dataset evaluated in the current study consisted entirely of simulated images which combined PET data on different scanners. In future studies a larger dataset of real patient images should be assessed. Fourth, we chose not to focus on the diagnosis of other types of dementia; e.g. fronto-temporal lobar degeneration, dementia with Lewy bodies and vascular dementia, which is sometimes difficult to distinguish from AD in a clinical setting. Fifth, the outcome measure was derived from a single software tool, and visual assessment by an experienced reader was not included. However, the main focus of this study was not on the tool itself but to assess the diagnostic accuracy of FDG PET from PET/MR with vendor-provided atlas AC for AD, in a controlled manner. This goal was fulfilled by the current study, because the diagnostic concept is similar between software or visual assessment. Sixth, we only evaluated a single atlas-based MRAC method. Cross validation using several MRAC methods, provided by other vendors or researchers, should be performed in the future.

## Conclusion

FDG-PET/MR based on atlas-based MR attenuation showed similar diagnostic accuracy than the CT-based method for the diagnosis of Alzheimer's disease and the prediction of the progression of mild cognitive impairment to Alzheimer's disease using the PMOD-based PALZ software, although with a minor reduction in sensitivity.

## Supporting information

**S1 Fig. Overview of the automated Alzheimer's discrimination software (PMOD PALZ) algorithm.** A detailed description is provided in the materials and methods section. (TIF)

**S2 Fig.** The original PET images extracted from ADNI-cohort (A) were spatially normalized to the SPM99 PET template, followed by smoothing with a Gaussian filter of 12 x 12 x 12 mm. Secondly, voxel values were normalized by dividing each image voxel value by the mean voxel value, averaged within a mask representing voxels with AD-preserved activity (B). The voxel-wise differences between the expected and patient-specific images are used to calculate t-values (C). The error map was obtained by dividing PET based on Atlas-AC by that on CT-AC, followed by the normalization to the identical space as spatially normalized ADNI-PET data. The original error map is shown in figures (D) and (F). We multiplied the normalized PET (B) with each normalized error map (D and F), followed by calculation of the t-value map of the simulated PET (not shown in this figure). The error of the t-value maps was obtained by subtracting the original t-value map from the simulated t-value map (E and G). The yellow VOI corresponds to AD-related voxels. The original PET-score of ADNI-data was 0.9385. The PET-score of simulated data was 1.0478 for pt-11 and 0.9497 for pt-14. More detailed figures are shown in S2 Fig.
(JPG)

## Acknowledgments

Data used in preparation of this article were obtained from the Alzheimer's Disease Neuroimaging Initiative (ADNI) database (adni.loni.usc.edu). As such, the investigators within the ADNI contributed to the design and implementation of ADNI and/or provided data but did not participate in analysis or writing of this report. A complete listing of ADNI investigators can be found at: http://adni.loni.usc.edu/wp-content/uploads/how_to_apply/ADNI_Acknowledgement_List.pdf. The primary investigator is Michael W. Weiner, MD. His contact email address is Michael.Weiner@ucsf.edu.

## Author Contributions

**Conceptualization:** Tetsuro Sekine, Gaspar Delso, Bradley Kemp, Edwin E. G. W. ter Voert, Martin Huellner, Patrick Veit-Haibach, Geoffrey Warnock.

**Data curation:** Tetsuro Sekine, Alfred Buck, Bradley Kemp, Edwin E. G. W. ter Voert, Martin Huellner, Patrick Veit-Haibach, Geoffrey Warnock.

**Formal analysis:** Tetsuro Sekine, Alfred Buck, Bradley Kemp, Edwin E. G. W. ter Voert, Martin Huellner, Patrick Veit-Haibach, Geoffrey Warnock.

**Funding acquisition:** Tetsuro Sekine.

**Investigation:** Tetsuro Sekine, Alfred Buck, Bradley Kemp, Edwin E. G. W. ter Voert, Martin Huellner, Patrick Veit-Haibach, Geoffrey Warnock.

**Methodology:** Tetsuro Sekine, Alfred Buck, Gaspar Delso, Bradley Kemp, Edwin E. G. W. ter Voert, Martin Huellner, Patrick Veit-Haibach, Geoffrey Warnock.

**Project administration:** Tetsuro Sekine, Alfred Buck, Gaspar Delso, Bradley Kemp, Edwin E. G. W. ter Voert, Martin Huellner, Patrick Veit-Haibach, Geoffrey Warnock.

**Resources:** Tetsuro Sekine, Gaspar Delso, Bradley Kemp, Edwin E. G. W. ter Voert, Martin Huellner, Patrick Veit-Haibach, Sandeep Kaushik, Florian Wiesinger, Geoffrey Warnock.

**Software:** Tetsuro Sekine, Gaspar Delso, Bradley Kemp, Edwin E. G. W. ter Voert, Martin Huellner, Patrick Veit-Haibach, Sandeep Kaushik, Florian Wiesinger, Geoffrey Warnock.

**Supervision:** Tetsuro Sekine, Gaspar Delso, Bradley Kemp, Edwin E. G. W. ter Voert, Martin Huellner, Patrick Veit-Haibach, Sandeep Kaushik, Florian Wiesinger, Geoffrey Warnock.

**Validation:** Tetsuro Sekine, Gaspar Delso, Bradley Kemp, Edwin E. G. W. ter Voert, Martin Huellner, Patrick Veit-Haibach, Geoffrey Warnock.

**Visualization:** Tetsuro Sekine, Gaspar Delso, Edwin E. G. W. ter Voert, Martin Huellner, Patrick Veit-Haibach, Geoffrey Warnock.

**Writing – original draft:** Tetsuro Sekine, Gaspar Delso, Geoffrey Warnock.

**Writing – review & editing:** Tetsuro Sekine, Gaspar Delso, Geoffrey Warnock.

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
