## [Decision Letter · Decision Letter 0]

27 Feb 2020

PONE-D-19-22678

The impact of atlas-based MR attenuation correction on the diagnosis of FDG-PET/MR for Alzheimer’s diseases— a simulation study combining multi-center data and ADNI-data

PLOS ONE

Dear Dr. Sekine,

Thank you for submitting your manuscript to PLOS ONE. After careful consideration, we feel that it has merit but does not fully meet PLOS ONE’s publication criteria as it currently stands. Therefore, we invite you to submit a revised version of the manuscript that addresses the points raised during the review process.

We would appreciate receiving your revised manuscript by Apr 11 2020 11:59PM. To enhance the reproducibility of your results, we recommend that if applicable you deposit your laboratory protocols in protocols.io, where a protocol can be assigned its own identifier (DOI) such that it can be cited independently in the future. For instructions see: http://journals.plos.org/plosone/s/submission-guidelines#loc-laboratory-protocols

We look forward to receiving your revised manuscript.

Kind regards,

Thomas Pyka

Academic Editor

PLOS ONE

Additional Editor Comments (if provided):

Dear Dr Sekine

The manuscript is interesting in showing the impact of MRAC on final diagnoses and I would be glad to receive a revised manuscript. Reviewer 1's decision was reject; however I propose you handle his comments as suggestions for a major revision. I think his comments on image coregistration should not be taken lightly though.

Best regards,

Thomas Pyka

Journal Requirements:

2. Our internal editors have looked over your manuscript and determined that it may be within the scope of our Early Diagnosis and Treatment of Alzheimer's Disease Call for Papers. This collection of papers is headed by a team of Guest Editors for PLOS ONE: Michael Weiner, Roberta Brinton, Jussi Tohka and Yona Levites. With this Collection we hope to bring together researchers working on a wide range of disciplines, from molecular and preclinical work, through to patient-centered studies, including clinical trials.   Additional information can be found on our announcement page: https://collections.plos.org/s/alzheimersdisease. If you would like your manuscript to be considered for this collection, please let us know in your cover letter and we will ensure that your paper is treated as if you were responding to this call. Agreeing to be part of the call-for-papers will not affect the date your manuscript is published. If you would prefer to remove your manuscript from collection consideration, please specify this in the cover letter.

http://jnm.snmjournals.org/content/57/12/1927.long

In your revision ensure you cite all your sources (including your own works), and quote or rephrase any duplicated text outside the methods section. Further consideration is dependent on these concerns being addressed.

4. Please ensure that you have included detailed inclusion and exclusion criteria for participants in your study (including both ADNI participants and participants you recruited yourself).

5. Thank you for including your ethics statement:  "This study was approved by each local institutional review board. All subjects provided signed informed consent prior to the examinations. All experiments were performed in accordance with relevant guidelines and regulations."

a.Please amend your current ethics statement to include the full name of the ethics committee/institutional review board(s) that approved your specific study.

b.Once you have amended this/these statement(s) in the Methods section of the manuscript, please add the same text to the “Ethics Statement” field of the submission form (via “Edit Submission”).

6. One of the noted authors is a group or consortium "Alzheimer’s Disease Neuroimaging Initiative". In addition to naming the author group, please list the individual authors and affiliations within this group in the acknowledgments section of your manuscript. Please also indicate clearly a lead author for this group along with a contact email address.

7. Thank you for stating the following in the Acknowledgments Section of your manuscript:

"This work was supported by JSPS KAKENHI (Grant Number 17K18160), Kurata Grants from the Hitachi Global Foundation (Grant Number 1309), research grants from Fukuda Foundation for Medical Technology and Terumo foundation for life sciences and arts. GW was funded by the Clinical Research Priority Program for Molecular Imaging of the University of Zurich (MINZ). Data collection and sharing for this project was funded by the Alzheimer's Disease Neuroimaging Initiative (ADNI) (National Institutes of Health Grant U01 AG024904) and DOD ADNI (Department of Defense award number W81XWH-12-2-0012)."

"The authors received no specific funding for this work."

8. Thank you for stating the following in the Acknowledgments Section of your manuscript:

"ADNI is funded by the National Institute on Aging, the National Institute of Biomedical Imaging and Bioengineering, and through generous contributions from the following: AbbVie, Alzheimer’s Association; Alzheimer’s Drug Discovery Foundation; Araclon Biotech; BioClinica, Inc.; Biogen; Bristol-Myers Squibb Company; CereSpir, Inc.; Cogstate; Eisai Inc.; Elan Pharmaceuticals, Inc.; Eli Lilly and Company; EuroImmun; F. Hoffmann-La Roche Ltd and its affiliated company Genentech, Inc.; Fujirebio; GE

Healthcare; IXICO Ltd.; Janssen Alzheimer Immunotherapy Research & Development, LLC.; Johnson & Johnson Pharmaceutical Research & Development LLC.; Lumosity; Lundbeck; Merck & Co., Inc.; Meso Scale Diagnostics, LLC.; NeuroRx Research; Neurotrack Technologies; Novartis Pharmaceuticals Corporation; Pfizer Inc.; Piramal Imaging; Servier; Takeda Pharmaceutical Company; and Transition Therapeutics. The Canadian Institutes of Health Research is providing funds to support ADNI clinical sites

in Canada. Private sector contributions are facilitated by the Foundation for the National Institutes of Health (www.fnih.org). The grantee organization is the Northern California Institute for Research and Education, and the study is coordinated by the Alzheimer’s Therapeutic Research Institute at the University of Southern California. ADNI data are disseminated by the Laboratory for Neuro Imaging at the University of Southern California."

We note that you have provided funding information that is not currently declared in your Competing Interests Statement. However, funding information should not appear in the Acknowledgments section or other areas of your manuscript. We will only publish funding information present in the Funding Statement/Competing Interests section of the online submission form.

Please remove any funding-related text from the manuscript and let us know how you would like to update your Competing Interests section. Currently, your Competing Interests reads as follows:

Please also confirm that this does not alter your adherence to all PLOS ONE policies on sharing data and materials, by including the following statement: "This does not alter our adherence to  PLOS ONE policies on sharing data and materials.” (as detailed online in our guide for authors http://journals.plos.org/plosone/s/competing-interests).  If there are restrictions on sharing of data and/or materials, please state these. Please note that we cannot proceed with consideration of your article until this information has been declared.

Reviewers' comments:

Reviewer's Responses to Questions

**Comments to the Author**

1. Is the manuscript technically sound, and do the data support the conclusions?

Reviewer #1: No

Reviewer #2: Partly

2. Has the statistical analysis been performed appropriately and rigorously? 

Reviewer #1: Yes

Reviewer #2: N/A

3. Have the authors made all data underlying the findings in their manuscript fully available?

Reviewer #1: No

Reviewer #2: No

4. Is the manuscript presented in an intelligible fashion and written in standard English?

Reviewer #1: Yes

Reviewer #2: Yes

5. Review Comments to the Author

Reviewer #1: PONE-D-19-22678

The manuscript presents the impact of a MRAC product on FDG PET/MR for the evaluation of Alzheimer’s disease (AD). The authors generated 47 PET error maps using the ratio of PET-MRAC/PET-CTAC and applied them to 203 Alzheimer’s Disease Neuroimaging Initiative (ADNI) cases after the identical normalization and smoothing. The results show that only sensitivity was slightly impaired in 5 out of 47 datasets for the discrimination of AD-patients from normal controls; however, the accuracy, sensitivity and specificity for predicting progression from mild cognitive impairment (MCI) to AD were not impaired. Thus, the study demonstrated that MRAC showed similar diagnostic accuracy to CTAC for the diagnosis of AD and the prediction of progression of MCI to AD. However, there are so many limitations in the study specifically in the data acquisition (PET data from PET/MR) and data processing (normalization and smoothing), which impairs the reliability of this study.

• The novelty of this study is on investigating the impact of MRAC on the diagnostic and prognostic accuracy of AD using PET images. However, since the impact of MRAC on PET images are very deeply investigated in a bunch of publications specifically for brain PET MRAC, the importance of this study is scientifically weak. Also, the diagnosis/prognosis is solely dependent on the commercial product (PMOD), which also reduces scientific interest towards this study.

• Also, the investigation was done through simulation, which impaired the reliability of this study because (1) spatial normalization and smoothing was applied to PET images for generating PET error maps (e.g., “To mitigate this, all images were smoothed with a Gaussian filter of 12 x 12 x 12 mm”), (2) PET data (from PET/MR) were not acquired in brain protocols (i.e., low quality as mentioned in the discussion) and (3) the error maps were applied to ADNI cases after normalization and smoothing. Considering the low resolution of PET images, the pre/post processing (e.g. spatial normalization to MNI space) can reduce or impair AD-associate information in final images.

• The authors do not demonstrate the accuracy of co-registration between CT and MR images, which is critical for the accuracy of generating PET error maps.

• The authors did not refer up-to-date publications regarding MRAC. Specifically, deep learning for directly converting MR to CT images has been well demonstrated in several publications.

1. Deep Learning MR Imaging-based Attenuation Correction for PET/MR Imaging. (2017)

2. Attenuation correction for brain PET imaging using deep neural network based on Dixon and ZTE MR images (2017)

3. Etc.

• “Our goal was to evaluate the state-of-the-art vendor-provided atlas AC, and to use the same PET raw data in the generation of error maps.” I can’t agree with this point that the atlas-based AC is the state-of-the-art.

• “We used a fully automated, commercially available, Alzheimer’s discrimination analysis to calculate a predictive value for AD”. If this manuscript is based on the partnership with the commercial product (PMOD), it should be clarified as a conflict-of-interest.

• There are too many limitations (as stated in the discussion) to consider that the results are scientifically reliable.

Reviewer #2: The author presented a study, to investigate the impact of atlas-based MR attenuation correction on the diagnosis of FDG-PET/MR for Alzheimer’s diseases using simulation data in manner of multi-center study. However, some critical problems exist in the methodology and the results should be addressed before considering for publication.

Introduction.

This part presented well, however I suggest to include one paragraph on more recent approaches on PET attenuation and scatter correction such as Deep Learning base methods.

Arabi, H., Zeng, G., Zheng, G., & Zaidi, H. (2019). Novel adversarial semantic structure deep learning for MRI-guided attenuation correction in brain PET/MRI. European journal of nuclear medicine and molecular imaging, 46(13), 2746-2759.

Shiri, Isaac, et al. "Direct attenuation correction of brain PET images using only emission data via a deep convolutional encoder-decoder (Deep-DAC)." European radiology 29.12 (2019): 6867-6879.

Sgard, Brian, et al. "ZTE MR-based attenuation correction in brain FDG-PET/MR: performance in patients with cognitive impairment." European radiology (2019): 1-10.

Methods:

The methods part is not provided very well and can’t be easily follow, I suggest to provide a flowchart and show the steps on this paper.

Please provide patients and demographic and clinical characteristics for each data sets.

Major comments:

Authors mentioned that, they enrolled patients with brain tumors, brain metastasis for generating the error maps. The propose methods was for AD, however the error map which provided for simulation data were provided on cancer and metastasis patients. These two dataset are totally different in nature and also the pattern. So, in my opinion the cancer data set is not suitable to provide the error map then apply on AD. I highly recommend to provide the error map using Normal or AD patients, because of different uptake, different distribution and pattern of brain.

Data for generation the error map (for simulation purpose) included the PSF and TOF information, dose the ADNI dataset included these information? The TOF information provided very useful information which help the attenuation correction in PET, if the ADNI doesn’t included the TOF information, then the generated maps shouldn’t also.

Providing error map using TOF dataset and then applying on non-TOF would be one of the main challenges, the authors should produce error map from non-TOF error map or apply the error map on AD images with TOF.

I would like to ask the authors to provide the error (i.e relative error) of each regions (respect to ground truth) using pmod atlas (i.e the Hammers maximum probability atlas), for comparing the CTAC and MRAC to show how the error map effect on datasets, rather than comparing on diagnostic analysis. Please provide the error of simulated datasets respect to ground truth.

Comparing the methods only on final results without showing what happened in process is not sufficient as the final results is depend on many parameters, then it’s not possible to explain why it does work or not working.

Results

Results part is to short, please explain more about the results, i.e how the map effect on datasets. An average error of either whole brain can assess the physical performance of the method. However, It possible that the errors of the each region (using atlas in PMOD) which is more important in AD diagnosis are higher than the whole brain error. The authors should be aware of this issue and then report the error of important regions in AD (to compare two methods)

Discussion

Authors provided the limitation of present study very well, I’m wondering about the dataset acquired with 2min which can lead to main issue in quantification (it’s almost low dose PET which is not clinically applicable in AD), I suggest to remove these data and re-evaluate or bring some background or justification for these data.

6. PLOS authors have the option to publish the peer review history of their article (what does this mean?). If published, this will include your full peer review and any attached files.

Reviewer #1: No

Reviewer #2: No

---

## [Author Response · Author response to Decision Letter 0]

14 Apr 2020

5. Review Comments to the Author

Reviewer #1: PONE-D-19-22678

The manuscript presents the impact of a MRAC product on FDG PET/MR for the evaluation of Alzheimer’s disease (AD). The authors generated 47 PET error maps using the ratio of PET-MRAC/PET-CTAC and applied them to 203 Alzheimer’s Disease Neuroimaging Initiative (ADNI) cases after the identical normalization and smoothing. The results show that only sensitivity was slightly impaired in 5 out of 47 datasets for the discrimination of AD-patients from normal controls; however, the accuracy, sensitivity and specificity for predicting progression from mild cognitive impairment (MCI) to AD were not impaired. Thus, the study demonstrated that MRAC showed similar diagnostic accuracy to CTAC for the diagnosis of AD and the prediction of progression of MCI to AD. However, there are so many limitations in the study specifically in the data acquisition (PET data from PET/MR) and data processing (normalization and smoothing), which impairs the reliability of this study.

Thank you for your many valuable comments.

• The novelty of this study is on investigating the impact of MRAC on the diagnostic and prognostic accuracy of AD using PET images. However, since the impact of MRAC on PET images are very deeply investigated in a bunch of publications specifically for brain PET MRAC, the importance of this study is scientifically weak. 

As the reviewer pointed out, there have been vast of published MRAC studies some of which included dementia patients. However, as described in the discussion part, there is very few or almost no paper to focus on the diagnostic accuracy of AD using commercially available MRAC method. From this point of view, we think that our study has substantial scientific merit.

Also, the diagnosis/prognosis is solely dependent on the commercial product (PMOD), which also reduces scientific interest towards this study.

The main focus of this study was not the performance of sole program itself. As described in the limitation section, the diagnostic concept is similar between PALZ and other software or even in visual assessment. PALZ package is standard analysis protocol which has been widely validated in large cohort study (e.g. ADNI, NEST-DD and SEAD-Japan) and used in clinical setting.

• Also, the investigation was done through simulation, which impaired the reliability of this study because (1) spatial normalization and smoothing was applied to PET images for generating PET error maps (e.g., “To mitigate this, all images were smoothed with a Gaussian filter of 12 x 12 x 12 mm”), (2) PET data (from PET/MR) were not acquired in brain protocols (i.e., low quality as mentioned in the discussion) and (3) the error maps were applied to ADNI cases after normalization and smoothing. Considering the low resolution of PET images, the pre/post processing (e.g. spatial normalization to MNI space) can reduce or impair AD-associate information in final images.

(1) This spatial normalization is one of the main steps in PALZ protocol. This step was not tailored for error map generation in the current study. The main purpose of this spatial normalization to error map was to generate simulation map combining ADNI data and real patients’ data in the same spatially normalized space. To clarify it, we added the sentence as below. “To mitigate this, all images were smoothed with a Gaussian filter of 12 x 12 x 12 mm which step has been included in PALZ as one of the analysis steps.”

(2) More than half of patients (27 patients at InB) in this study underwent 10-minutes PET scan which was expected to have sufficient image quality on high-sensitivity PET/MR system with SiPM detector. We proved that the difference of PET score error between InA and InB was not significantly different (-0.070±0.045 vs. -0.066±0.046; p=0.392). We have added this statement in the limitation section. In addition, if the 2min acquisition leads to substantial error even after applying large Gaussian filter, the PET score difference was expected to be under or overestimated than the ideal situation. It means that our main result that Atlas-based MRAC is accurate would be maintained even in the ideal situation.

(3) As described above, normalization and smoothing is one of the main steps of general PALZ analysis steps. The main purpose of this step is to compensate the difference of SNR or spatial resolution among institutions. The diagnostic accuracy of original ADNI data is maintained after this step (83.2%) as shown in the result section.

• The authors do not demonstrate the accuracy of co-registration between CT and MR images, which is critical for the accuracy of generating PET error maps.

We used a standard method of co-registration between two modalities, mutual information matching algorithm by using widely available software (Pmod). We agree that the registration error may overestimate the error from MRAC. However, this overestimation has subtle impact on our main conclusion that MRAC can maintain the diagnostic accuracy for the diagnosis of AD. If the reviewer suggests, we will add this topic to the limitation section. 

• The authors did not refer up-to-date publications regarding MRAC. Specifically, deep learning for directly converting MR to CT images has been well demonstrated in several publications.

1. Deep Learning MR Imaging-based Attenuation Correction for PET/MR Imaging. (2017)

2. Attenuation correction for brain PET imaging using deep neural network based on Dixon and ZTE MR images (2017)

3. Etc.

We have added some references regarding deep-learning-based MRAC. Thank you.

• “Our goal was to evaluate the state-of-the-art vendor-provided atlas AC, and to use the same PET raw data in the generation of error maps.” I can’t agree with this point that the atlas-based AC is the state-of-the-art.

We agree that Atlas-MRAC is no longer state-of-the-art method. We have deleted “state-of-the-art” method.

• “We used a fully automated, commercially available, Alzheimer’s discrimination analysis to calculate a predictive value for AD”. If this manuscript is based on the partnership with the commercial product (PMOD), it should be clarified as a conflict-of-interest.

We used only commercially available functions on Pmod. We already stated as “Automated AD probability assessment was performed in a commercially available tool (PMOD Alzheimer’s Discrimination, PALZ).”. If the reviewer suggested that we will add some more statements.

• There are too many limitations (as stated in the discussion) to consider that the results are scientifically reliable.

We agree that there are too many limitations. However, each limitation is not so critical.

[1] Some of PET data for error map generation was done from 2min brain PET scan. However, we proved that 2min brain PET scan didn’t overestimate/underestimate the final result compared to 10 min brain PET scan.

[2] PET data was taken from oncologic patients. However, MRAC error was mainly derived from the head skull which does not vary among oncologic patient and dementia subjects.

[3] We stated the sample size is small. To be honest, n=47 is not critically small in this kind of study.

[4] We only assessed AD-related diagnostic accuracy. To be honest, this setup is a common study design in the diagnostic accuracy of dementia.

[5] Outcome measure was done by PALZ software. The diagnostic concept in PALZ software is similar in other software or visual assessment. The detection of AD-related uptake change is the key diagnostic process. We think that the method in the current study is relatively objective because whole process was done semi-automatically.

[6] We validated the sole MRAC method, Atlas-AC. This method is one of the main commercially available attenuation correction methods on GE-PET/MR which is widely used in the clinical setting. 

Reviewer #2: The author presented a study, to investigate the impact of atlas-based MR attenuation correction on the diagnosis of FDG-PET/MR for Alzheimer’s diseases using simulation data in manner of multi-center study. However, some critical problems exist in the methodology and the results should be addressed before considering for publication.

Thank you for your many valuable comments.

Introduction.

This part presented well, however I suggest to include one paragraph on more recent approaches on PET attenuation and scatter correction such as Deep Learning base methods.

Arabi, H., Zeng, G., Zheng, G., & Zaidi, H. (2019). Novel adversarial semantic structure deep learning for MRI-guided attenuation correction in brain PET/MRI. European journal of nuclear medicine and molecular imaging, 46(13), 2746-2759.

Shiri, Isaac, et al. "Direct attenuation correction of brain PET images using only emission data via a deep convolutional encoder-decoder (Deep-DAC)." European radiology 29.12 (2019): 6867-6879.

Sgard, Brian, et al. "ZTE MR-based attenuation correction in brain FDG-PET/MR: performance in patients with cognitive impairment." European radiology (2019): 1-10.

We have added suggested references. Thank you.

Methods:

The methods part is not provided very well and can’t be easily follow, I suggest to provide a flowchart and show the steps on this paper.

We had already provided the flow chart of each process in figure 1 and supplemental figure 1. To improve readability, we have added the statement of referred figure in the first paragraph of materials and methods section where the summary of whole steps was described.

Please provide patients and demographic and clinical characteristics for each data sets.

We have added the detailed information for each dataset.

Major comments:

Authors mentioned that, they enrolled patients with brain tumors, brain metastasis for generating the error maps. The propose methods was for AD, however the error map which provided for simulation data were provided on cancer and metastasis patients. These two dataset are totally different in nature and also the pattern. So, in my opinion the cancer data set is not suitable to provide the error map then apply on AD. I highly recommend to provide the error map using Normal or AD patients, because of different uptake, different distribution and pattern of brain.

We expected that the morphology without atrophy, and specific FDG-uptake distribution of oncology patients may not induce a large effect on the FDG-uptake error led by MRAC. The error is mainly caused by the error of the skull bone estimation by atlas-MRAC. This topic was already mentioned in the limitation part as “the error introduced by MRAC primarily results from differences in skull bone detection which is not expected to systematically vary among oncology and AD patients.”. 

Data for generation the error map (for simulation purpose) included the PSF and TOF information, dose the ADNI dataset included these information? The TOF information provided very useful information which help the attenuation correction in PET, if the ADNI doesn’t included the TOF information, then the generated maps shouldn’t also.

Providing error map using TOF dataset and then applying on non-TOF would be one of the main challenges, the authors should produce error map from non-TOF error map or apply the error map on AD images with TOF.

The main goal of this study was to combine a well-controlled dataset and clinically realistic error map. There is no clinical situation that MRAC without TOF reconstruction on GE PET/MR scanner because TOF reconstruction reduces the error from MRAC. Ideally, well-controlled PET data with TOF reconstruction should be used. However, such open-data is not accessible yet. 

We agree that this point is the main drawback of our simulation study. To clarify it, we have added the comment in the limitation part as “In addition, the dataset evaluated in the current study consisted entirely of simulated images which combined PET data on different scanner.”

I would like to ask the authors to provide the error (i.e relative error) of each regions (respect to ground truth) using pmod atlas (i.e the Hammers maximum probability atlas), for comparing the CTAC and MRAC to show how the error map effect on datasets, rather than comparing on diagnostic analysis. Please provide the error of simulated datasets respect to ground truth.

Comparing the methods only on final results without showing what happened in process is not sufficient as the final results is depend on many parameters, then it’s not possible to explain why it does work or not working.

We already showed that detail of MRAC error in other papers (Delso G. Neuroimage 2018;181:403-413. and Sekine T. J Nucl Med 2016;57:1927-1932.).

To avoid the duplication, we only added the evaluation about the error distribution.

In “Evaluation of diagnostic accuracy of Alzheimer’s disease” we have amended as below.

“First, to clarify the distribution of MRAC error, we calculated the averaged error in whole SPM 99 PET voxels, whole AD-related voxels and whole non-AD related voxels in each of the 47 normalized error map ((_Error^Norm)〖PET〗_ ^(pt-i)).”

In result section, we have amended as below

“In 47 normalized error map, whole voxels of SPM 99 were slightly underestimated (-1.37%±1.98%). In detail, AD-related voxels were less underestimated than non AD-related voxels (-0.86%±2.03% vs. -1.59%±1.98%).”

Results

Results part is to short, please explain more about the results, i.e how the map effect on datasets. An average error of either whole brain can assess the physical performance of the method. However, It possible that the errors of the each region (using atlas in PMOD) which is more important in AD diagnosis are higher than the whole brain error. The authors should be aware of this issue and then report the error of important regions in AD (to compare two methods)

As described above, we added the evaluation of the error distribution caused by MRAC. The additional analysis shows that AD-related voxels were less underestimated (-0.86%) than non AD-related voxels (-1.59%). It means that AD-related voxels were overestimated after value-normalization, resulting in decreasing t-value. It is in line with the result that PET score was slightly underestimated (-0.068).

Discussion

Authors provided the limitation of present study very well, I’m wondering about the dataset acquired with 2min which can lead to main issue in quantification (it’s almost low dose PET which is not clinically applicable in AD), I suggest to remove these data and re-evaluate or bring some background or justification for these data.

More than half of patients (27 patients at InB) in this study underwent 10-minutes PET scan which was expected to have sufficient image quality on high-sensitivity PET/MR system with SiPM detector. We proved that the difference of PET score error between InA and InB was not significantly different (-0.070±0.045 vs. -0.066±0.046; p=0.392). We have added this statement in the limitation section. In addition, if the 2min acquisition leads to substantial error even after applying large Gaussian filter, the PET score difference was expected to be under- or overestimated than the ideal situation. It means that our main result that Atlas-based MRAC is accurate would be maintained even in the ideal situation.

---

## [Decision Letter · Decision Letter 1]

15 May 2020

The impact of atlas-based MR attenuation correction on the diagnosis of FDG-PET/MR for Alzheimer’s diseases— a simulation study combining multi-center data and ADNI-data

PONE-D-19-22678R1

Dear Dr. Sekine,

We are pleased to inform you that your manuscript has been judged scientifically suitable for publication and will be formally accepted for publication once it complies with all outstanding technical requirements.

With kind regards,

Thomas Pyka

Academic Editor

PLOS ONE

Additional Editor Comments (optional):

Reviewers' comments:

Reviewer's Responses to Questions

**Comments to the Author**

1. If the authors have adequately addressed your comments raised in a previous round of review and you feel that this manuscript is now acceptable for publication, you may indicate that here to bypass the “Comments to the Author” section, enter your conflict of interest statement in the “Confidential to Editor” section, and submit your "Accept" recommendation.

Reviewer #2: All comments have been addressed

2. Is the manuscript technically sound, and do the data support the conclusions?

Reviewer #2: Partly

3. Has the statistical analysis been performed appropriately and rigorously? 

Reviewer #2: Yes

4. Have the authors made all data underlying the findings in their manuscript fully available?

Reviewer #2: Yes

5. Is the manuscript presented in an intelligible fashion and written in standard English?

Reviewer #2: Yes

6. Review Comments to the Author

Reviewer #2: (No Response)

7. PLOS authors have the option to publish the peer review history of their article (what does this mean?). If published, this will include your full peer review and any attached files.

Reviewer #2: No

---

## [Editor Report · Acceptance letter]

20 May 2020

PONE-D-19-22678R1 

The impact of atlas-based MR attenuation correction on the diagnosis of FDG-PET/MR for Alzheimer’s diseases— a simulation study combining multi-center data and ADNI-data 

Dear Dr. Sekine:

I am pleased to inform you that your manuscript has been deemed suitable for publication in PLOS ONE. Congratulations! Your manuscript is now with our production department. 

With kind regards,

on behalf of

Dr. Thomas Pyka 

Academic Editor

PLOS ONE